# Gender Moderates the Associations Between Responsiveness to Alarming Oral Sensations, Depressive Symptoms, and Dietary Habits in Adolescents

**DOI:** 10.3390/nu17101653

**Published:** 2025-05-12

**Authors:** Leonardo Menghi, Lara Fontana, Silvia Camarda, Isabella Endrizzi, Maria Pina Concas, Paolo Gasparini, Flavia Gasperi

**Affiliations:** 1Center Agriculture Food Environment, University of Trento, Via Mach 1, 38098 San Michele all’Adige, Italy; flavia.gasperi@unitn.it; 2Department of Green Technology, University of Southern Denmark, Campusvej 55, 5230 Odense, Denmark; 3Research and Innovation Centre, Edmund Mach Foundation, Via Mach 1, 38098 San Michele all’Adige, Italy; lara.fontana@fmach.it (L.F.); isabella.endrizzi@fmach.it (I.E.); 4Department of Medicine, Surgery and Health Sciences, University of Trieste, Strada di Fiume 447, 34149 Trieste, Italy; silvia.camarda@phd.units.it (S.C.); paolo.gasparini@burlo.trieste.it (P.G.); 5Institute for Maternal and Child Health, I.R.C.C.S. “Burlo Garofolo”, Via dell’Istria, 65/1, 34137 Trieste, Italy; mariapina.concas@burlo.trieste.it

**Keywords:** adolescents, gender, depressive symptoms, taste, oral responsiveness, diet

## Abstract

**Background/Objectives**: As a peripheral effect of depression-related traits, sensory responses may predispose individuals to depressive symptoms by prompting suboptimal dietary patterns with long-term effects on mood. Mood disturbances in adolescence are strong predictors of adult mental illness, making it crucial to identify factors that may shift transient mood fluctuations into more severe mental health issues during this vulnerable period. Given the substantial gender differences in susceptibility to comorbidities of depression, we examined whether the link between sensory perception and depressive symptoms in nonclinical adolescents varied by gender and was related to dietary habits. **Methods**: In this cross-sectional study, 232 healthy adolescents (41.8% girls, aged 13–17) reported their diet over the past year using the EPIC Food Frequency Questionnaire and rated their liking and perceived intensity of oral sensations from four grapefruit juices and dark chocolate puddings with varying sucrose levels. Additionally, participants completed assessments of anxiety, neuroticism, pickiness, body dissatisfaction, and the Patient Health Questionnaire (PHQ-9) to evaluate depressive symptoms. **Results**: We found that girls exhibited higher levels of depression, anxiety, neuroticism, and pickiness compared to boys (Wilcoxon Rank Sum Test), and that greater responsiveness to bitterness (e.g., β = 0.264, *p* = 0.037) and astringency (β = 0.269, *p* = 0.029) predicted higher depressive symptoms exclusively in girls. PHQ-9 scores were positively associated with alcohol use in both girls (ρ = 0.176, *p* = 0.003) and boys (ρ = 0.148, *p* = 0.004) and inversely related to the intake of beneficial nutrients (e.g., fiber, polyunsaturated fats), particularly in girls. Intriguingly, moderation analyses suggested that associations between nutrient intake and acuity for alarming oral sensations were largely moderated by depression-related traits in girls, but not in boys. **Conclusions**: Our findings suggest that gender moderates the links between depressive symptoms, sensory perception, and dietary habits in healthy adolescents, possibly reflecting gender-specific coping strategies for comorbidities of depression.

## 1. Introduction

Adolescence is a transient stage of life marked by profound physiological and psychosocial changes that can increase vulnerability to mood disorders. As emotional challenges intensify, mood disturbances may escalate to more significant mental health conditions, including depression. Characterized by persistent loss of pleasure or interest in activities and depressed mood [1], depressive disorders are one of the leading causes of morbidity globally among adolescents, contributing to 15% of the disease burden in this age group [2]. Approximately 34% of adolescents aged 10–19 are thought to experience intense depressive symptoms [3], with prevalence doubling during the first year of the COVID-19 pandemic [4] and increasing by 14% from the first (24%) to the second (38%) decade of the 21st century [3]. In particular, girls are twice as likely as boys to experience severe depressive symptoms, e.g., [5,6,7]. This gap typically emerges during preadolescence and peaks at ages 15–18, e.g., [6], likely due to factors such as earlier onset of puberty, increased exposure to severe stressors, intensified societal and academic pressures, and heightened susceptibility to the psychosocial correlates of depression (for review, see [7]). This is inevitably a cause for concern, as severe mood disturbances during adolescence are a strong predictor of mental illness in adulthood [8], and early symptoms often go undetected and untreated [2,9]. Therefore, identifying factors that may facilitate the progression from short-term mood fluctuations to significant mental health problems during adolescence is a critical public health priority.

In this context, changes in dietary habits are increasingly recognized as one of the most promising modifiable risk factors for preventing depression (for review, see [10]). In particular, an adequate intake of vegetables, fruits, vegetable oils, fish, and whole grains, along with limited consumption of refined grains, simple sugars, red and processed meats, and full-fat dairy products, has been consistently associated with a reduced risk of depression among adolescents worldwide, e.g., [11,12,13]. Indeed, diets rich in plant-based nutrients, such as the Mediterranean Diet, have been shown to reduce plasma levels of pro-inflammatory cytokines (e.g., IL-6, INF-γ, TNF-α), thereby counteracting biological mechanisms implicated in the etiology of depression, including systemic oxidative stress and dysregulation in the production of glucocorticoids, monoamines, and brain-derived neurotrophic factor (for review, see [14]). Therefore, uncovering barriers to adherence to dietary patterns beneficial for mood is one of the essential steps in addressing the rising rates of depression among youth.

Despite greater independence compared to childhood and significant familial and social influences, the sensory properties of food (taste, smell, flavor, texture, appearance) remain one of the key determinants of dietary choices during this developmental stage (for review, see [15]). Adolescents generally favor more intensely sweet tastes relative to adults [16] and tend to avoid foods that evoke inherently disliked oral sensations, such as bitterness [15]. However, there are substantial interpersonal differences in taste perception, which can influence adherence to or deviation from healthy dietary patterns, e.g., [17,18]. An example is the genetic ability to perceive bitter thiourea compounds, such as phenylthiocarbamide or 6-n-propylthiouracil (PROP), which can have downstream effects on food preferences (for review, see [18]). While PROP perception is a continuous trait, individuals are typically classified as non-tasters (NTs), medium-tasters (MTs), or super-tasters (STs) based on cut-offs that reflect varying levels of perceived bitterness, from null to extreme, with haplotypes in the *TAS2R38* gene largely responsible for this phenotypic variation [18,19].

In adolescents, PROP status appears to influence food liking more than actual intake. For instance, NTs rate cruciferous vegetables as more pleasant than STs [20], whereas STs find animal-based foods (e.g., bacon, fried chicken, and herring), sauces, condiments [21], and sweetened foods more appealing [22]. While these findings might suggest that STs are more likely to adopt suboptimal dietary patterns, previous reports yielded mixed results. Only one study reported a higher sugar intake among STs relative to NTs [22], while others failed to link PROP phenotypes and/or genotypes to diet [20,21,23]. Thus, although the evidence remains insufficient to draw definitive conclusions due to small sample sizes and varying methods of operationalizing PROP status and dietary habits [24], other factors underlying variations in taste perception may contribute more to undesired dietary patterns during this phase of life.

In addition to PROP, notable gender differences in taste perception have been documented across the lifespan [22,25]. Girls at various stages of adolescence generally show greater responsiveness to oral stimuli than boys, e.g., [22,26], and are more likely to be STs [19]. However, whether these differences are primarily psychological or physiological remains debated [27]. In fact, a large body of literature suggests that sensory perception can also be influenced by mood states and personality traits (for reviews, see [17,28]). In nonclinical adults, anxiety and exposure to stressors (e.g., airhorn sounds, cold pressor test) tend to express higher intensity ratings to sweet, salty, and bitter stimuli [29,30,31], and being neurotic [32] or a picky eater [33] has been positively associated with acuity for bitter and for sweet and bitter aqueous solutions, respectively.

Interestingly, similar findings have been reported for depression. Dess and Chapman [34] found a positive correlation between depressive symptoms and bitterness ratings of quinine solutions, and Platte et al. [31] extended the same findings to the sweet taste elicited by varying levels of sucrose in water. Nevertheless, a link between greater responsiveness to oral stimuli and pronounced internalizing symptoms (anxiety, depression, stress) has not always been supported [35,36,37], and the opposite has often been observed in clinically depressed adults (for review, see [28]). Although findings are inconclusive and largely based on small adult samples, there is evidence that vulnerability to correlates of depression may drive variations in taste perception, potentially shaping dietary choices with long-term negative effects on mood.

Building on this, adolescents worldwide often cite the unpleasant sensory properties of healthy foods as a key barrier to healthy eating, leading them to opt for less nutritious options [15]. This is evident because prototypical healthy foods (e.g., vegetables, fruits, nuts, vegetable oils), whether raw or cooked, can evoke a range of sensory qualities such as bitterness, sourness, or astringency (hereafter referred to as alarming oral sensations), which elicit psychobiological states (i.e., arousal) that enhance alertness and facilitate rejection due to their perceived dangerousness [38]. However, the impact of arousal on food choices varies widely among individuals and depends on how well a sensation aligns with their optimal level of activation [38]. Consequently, an individual’s susceptibility to sources of arousal in foods (stimulus intensity, novelty, and complexity) is contingent upon their responsiveness to exogenous stimuli and mood state (for review, see [39]). Individuals with traits that heighten responses to external inputs, such as those related to depression, may thus be more reactive to negative arousal from food and potentially reinforce undesired dietary choices with long-term effects on mood. Given the increased vulnerability of girls to depression and its comorbidities [3,7], this holds the potential to yield new insights into gender-based antecedents of depressive symptoms and warrants further investigation.

While taste plays a crucial role in shaping food habits among adolescents [15], and diet may prevent depressive disorders [10], no studies have yet simultaneously examined the links between sensory perception, depressive symptoms, and dietary habits in healthy adolescents, particularly concerning variations in vulnerability to correlates of depression. Additionally, the current knowledge has focused on either small adult samples, e.g., [31,34,35], or documented inconclusive results on the impact of taste on dietary habits due to a paucity of studies [24]. Also, the majority of previous research has relied on detection or recognition thresholds of diluted tastants in simple solutions, which have shown minimal correlations with everyday perceptions at suprathreshold levels [40]. To address this gap, the use of actual or model foods with varying tastant concentrations could offer more ecologically relevant insights, e.g., [41].

Against this backdrop, we examined whether the link between sensory perception and depressive symptoms in nonclinical adolescents varied by gender and was related to dietary habits. Additionally, we explored whether these associations were moderated by common psychosocial correlates of depression, such as anxiety [42], neuroticism [43], picky eating [33], and body dissatisfaction [44].

In this study, 232 nonclinical adolescents provided data on PROP responsiveness, as well as hedonic and psychophysical ratings to oral stimuli elicited by four variants of two food models with varying sucrose levels. Participants also completed the European Prospective Investigation into Cancer and Nutrition Food Frequency Questionnaire (EPIC-FFQ) [45] to monitor their habitual diet, a series of questionnaires assessing the aforementioned correlates of depression, and reported their depressive symptoms over the previous two weeks using the nine-item Patient Health Questionnaire (PHQ-9) [46].

## 2. Materials and Methods

### 2.1. Participants

This work builds on a broader project aimed at elucidating the genetic, non-genetic, and psychosocial determinants of individual variations in sensory perception among adolescents and their impact on diet. Due to the lack of comparable studies, sample size was estimated based on a previous study from our group involving healthy young adults (aged 18–30) from the same location (Autonomous Province of Trento, Italy) and engaged in similar experimental tasks [47]. In that study, we found significant differences (Cohen’s d = 0.402) in responsiveness to bitterness and sourness elicited by commercially available foods between groups with comparable salivary microbial profiles. Accordingly, we conducted a power analysis using the pwr.t.test R function [48], which indicated that 198 participants would be needed to detect such an effect with 80% power at α = 0.05 (two-tailed). To account for deviations from normality and potential dropouts, the target sample was increased by 15%, resulting in an expected sample size of 228 participants.

This was later accommodated with a final sample of 232 healthy adolescents (41.8% girls, 13–17 years; mean age ± SD = 14.5 ± 0.6 years; mean BMI ± SD = 21.0 ± 3.2 kg/m^2^), who were recruited via a series of promotional events targeting both potential participants and their parents from two high schools in the Autonomous Province of Trento, Italy.

No formal exclusion criteria were applied. Nonetheless, participants reported no current diagnosis of major depressive disorder, nor were they undergoing treatment or had used medications in the past 6 months that could affect mood or taste function. Furthermore, girls and boys did not differ (*p* > 0.05) in terms of age, BMI, smoking habits, weekly alcohol consumption, food allergies, and habitual diet. In contrast, boys engaged in more physical activity than girls (t = 2.532, *p* = 0.012). A full demographic description of our sample is provided in Appendix A.

Before data collection, parents or legal guardians provided written consent, and adolescents gave written assent. The study was approved by the Research Ethics Committee of the University of Trento (n° prot. 2023-047, approved on 28 September 2023) and followed the principles of the Declaration of Helsinki (as amended in Fortaleza, Brazil, 2013).

### 2.2. Overview of Data Collection

To align with the project objectives, a 7-day cross-sectional protocol was designed to collect a broad range of sensory, psychometric, demographic, health-related, and dietary data, as well as biological samples (saliva and tongue swabs) for metagenomic and genetic analyses. Key variables used in this study are highlighted in bold in the graphical overview of the experimental design (Figure 1), while Table 1 and Table 2 provide details on tasks and measures used during data collection.

To optimize logistics, participants took part in two distinct work sessions: one remote and one conducted in an ISO 8589:2007 [49] compliant sensory lab. At least 7 days before the lab session, a researcher visited participants during school hours to provide detailed instructions for the home tasks (Table 1). These included an online questionnaire assessing demographics, anthropometrics, physical health status, lifestyle habits, stated preferences for a list of 57 food items, and selective/avoidant feeding attitudes using the Food Neophobia Scale and the Adult Picky Eating Questionnaire. Additionally, to assess their habitual diet, participants completed a paper version of the EPIC-FFQ, which they returned at the beginning of the lab session.

After the remote session, participants were instructed to refrain from eating, drinking (except water), smoking, and brushing their teeth for at least 2 h before visiting the sensory lab at the Edmund Mach Foundation (San Michele all’Adige, Trento, Italy). Due to the extensive data collection, the lab session was divided into four slots (Parts 1–4) to minimize participants’ burden and ensure data quality (Figure 1).

Part 1 (Table 2) began with the autonomous collection of a saliva sample and a tongue swab for metagenomic analyses. Participants then rated their familiarity with the same 57 food items assessed remotely and completed the Arnett Inventory of Sensation Seeking to evaluate their inclination for novel, varied, and intense experiences. An additional salivary sample was collected for genotyping before concluding Part 1 (Figure 1).

In Part 2 (Table 2), participants evaluated their liking of two independent sets of four variants each of two model foods (grapefruit juice, dark chocolate pudding) with varying sucrose concentrations. Sensory evaluations were interspersed with a 5 min break featuring logic games to maintain motivation. The following assessment of personality dimensions through the Goldberg’s Big Five Inventory marked the end of this slot.

Part 3 (Table 2) focused on psychophysical responses to oral sensations elicited by the same model foods from Part 2. Participants then completed the Screen for Child Anxiety Related Disorders and the Body Image Dimensional Assessment to assess their generalized, social, and school anxiety and estimate their level of body dissatisfaction, respectively.

In Part 4 (Table 2), participants rated the perceived bitterness of two aqueous solutions containing PROP and reported the severity of depressive symptoms experienced in the past 2 weeks through the Patient Health Questionnaire. Next, they were asked about their smoking habits, alcohol, snacks, and sweetened beverages consumption, and habitual use of social networks before concluding the study.

Each task slot (Parts 1–4) was separated by a break (B1–B3) of at least 15 min (Figure 1), during which detailed verbal and practical instructions for the upcoming tasks were provided. Data were collected using the EyeQuestion software (version 5.5.0, Elst, The Netherlands), except for the EPIC-FFQ, which was completed in paper form. Data collection took place between November and December 2023, with lab sessions held from 9:00 AM to 12:00 PM. The following sections provide details on the measures used to achieve the aims of the current work.

**Table 1 nutrients-17-01653-t001:** Data collected during the remote work session. Relevant questionnaires, response options, number of items, rating scales, and references are listed. ^†^ Data collected in paper form.

Questionnaire	Output	Options (Scale)	References
Demographics	Age	Years old at the moment of the test	
	Gender	Girl/Boy	
Anthropometrics	Weight	40–180 kg	
	Height	100–220 cm	
Physical health status (ongoing diagnosis or within the last 6 months)	Chronic diseases	Asthma; COVID-19; Type I or II diabetes; Celiac disease; Crohn’s disease; Gastroesophageal reflux; Hiatal hernia; Irritable bowel syndrome	
	Oral diseases	Aphthous ulcers; Halitosis; Oral candidiasis; Gingivitis; Periodontitis; Xerostomia	
	Psychiatric disorders	Anorexia nervosa; Bulimia nervosa; Binge-eating disorder; Generalized anxiety disorder; Major depressive disorder; Autism spectrum disorder	
	Taste and smell disorders	Ageusia; Anosmia; Hypogeusia; Hyposmia	
	Food allergies	Yes/No [if yes, details asked]	
	Food intolerances	Yes/No [if yes, details asked]	
Medications and supplements use (past 6 month)	Antibiotics use	Yes/No	
	Medicines use	Yes/No [if yes, details asked]	
	Probiotics use	Yes/No	
Lifestyle habits	Diet	10 items	[50]
	International Physical Activity Questionnaire (IPAQ)	7 items	[51]
	Oral Hygiene Behavior Questionnaire	8 items	[52]
Food Liking Questionnaire	Stated liking for a range of foods	57 items rated (9-point hedonic scale; 1 = extremely disliked, 9 = extremely liked)	Adapted from [53]
Food Neophobia Scale	Selective and avoidant feeding attitudes	10 items (7-point Likert scale; 1 = strongly disagree; 7 = strongly agree)	[54]
Adult Picky Eating Questionnaire		20 items (5-point Likert scale; 1 = never; 5 = always)	[55]
Dietary habits	EPIC Food Frequency Questionnaire ^†^	163 items	[45]

**Table 2 nutrients-17-01653-t002:** Data collected during the four task slots (Part) designed for the lab session. Relevant tasks, expected outputs, number of items, rating scales, and references are listed. Gj: grapefruit juice series; Cp: dark chocolate pudding series.

Part	Task	Output	Items	Scale	References
1	Saliva and tongue swab (metagenomics)	One salivary and one tongue swab sample for shotgun metagenomics			
	Food familiarity questionnaire	Familiarity with a range of foods	57 items	5-point Likert scale (1 = I do not recognize it; 5 = I regularly eat it)	Adapted from [53]
	Arnett Inventory of Sensation Seeking questionnaire	Inclination for novel, varied, and intense experiences	20 items	4-point Likert scale (1 = describes me very well; 4 = does not describe me at all)	[56]
	Saliva (genotyping)	One salivary sample for SNP-array			
2	Liking _Gj_	Liking ratings for the four variants of Gj		Labeled Affective Magnitude scale (0 = greatest imaginable dislike; 100 = greatest imaginable like)	[57]
	Liking _Cp_	Liking ratings for the four variants of Cp			
	Goldberg’s Big Five Inventory questionnaire	Personality traits	30 items	7-point Likert scale (1 = does not apply to me at all, 7 = applies to me very well)	[58]
3	Intensity _Gj_	Intensity ratings for oral sensations from the four Variants of Gj		generalized Labeled Magnitude Scale (0 = no sensation, 100 = the strongest imaginable sensation of any kind)	[59]
	Intensity _Cp_	Intensity ratings for oral sensations from the four variants of Cp			
	Screen for Child Anxiety Related Disorders questionnaire	Level of generalized, social, and school anxiety	17 items	3-point Likert scale (1 = almost never, 3 = often)	[60]
	Body Image Dimensional Assessment questionnaire	Level of body dissatisfaction	4 items	Line scale from 1.8 to 5.2	[61]
4	Intensity _PROP_	Intensity ratings from two PROP aqueous solutions		generalized Labeled Magnitude Scale (0 = no sensation, 100 = the strongest imaginable sensation of any kind)	[59]
	Patient Health Questionnaire	Severity of depressive symptoms	9 items	4-point Likert scale (0 = never; 3 = nearly every day)	[46]
	Food-related and lifestyle habits	Monthly frequency of smoking cigarettes and e-cigs	3 items	7-point Likert scale (1 = never; 7 = 30 days)	
		Weekly intake of beer, wine, liquors, and cocktails	4 items	7-point Likert scale (1 = never; 7 = more than once a day)	
		Weekly intake of snacks and sweetened beverages	14 items	7-point Likert scale (1 = never; 7 = more than once a day)	
		Daily use of social networks	2 items	5-point Likert scale (1 = less than 1 h; 5 = more than 4 h)	

### 2.3. Sensory Tasks

#### 2.3.1. Stimuli

We aimed to develop four variants of two food models that would elicit a range of oral sensations, including sweet, bitter, sour, astringent, and flavors (grapefruit and chocolate), with the potential to mimic the spectrum of intensities experienced in everyday foods. Given its strong acceptance at high intensities among adolescents [15], sweetness was selected as the target sensation for modulation. Each food model had to (a) be widely available on the Italian market and familiar to our target population; (b) evoke alarming oral sensations that could be masked by increasing sweetness levels while maintaining a congruent sensory profile; (c) be simple to prepare, store, portion, and consume at room temperature; (d) be accepted by omnivores, vegetarians, and vegans. As a result, commercially available 100% grapefruit juice (Gj) and dark chocolate pudding (Cp) base formulations [53] were deemed the most suitable options (Table 3).

For each food model, four different sucrose concentrations were initially chosen based on previous studies using Gj [41] and Cp [53], and expected to elicit an increasingly higher level of sweetness while progressively decreasing relevant alarming oral sensations. Given that these sucrose levels had been tested either in a different Gj base [41] or in adults (Cp) [53], an initial evaluation was conducted with a panel of 39 individuals (58.9% girls, mean age ± SD = 44.7 ± 14.5 y) experienced in sensory analysis (Pilot 1). Results (Appendix A) indicated that slight adaptations to the sucrose span in both Gj and Cp were necessary to more effectively differentiate each series of products. Following these adjustments, a group of adolescents (Pilot 2, n = 16; 26.6% girls, mean age ± SD = 17.3 ± 0.4 y) rated the perceived intensities of oral sensations evoked by the optimized food models (Table 3) using the generalized Labeled Magnitude Scale (gLMS, 0 = no sensation, 100 = the strongest imaginable sensation of any kind), which revealed clearer sensory differences among variants of both Gj and Cp (Appendix A). The suitability of both food models was later corroborated by our population (Part 3, Figure 1), as psychophysical responses (gLMS) to sweetness exhibited a systematic increase, while bitterness, sourness, and astringency decreased as sucrose concentration increased (Appendix A). For all testing, Gj was prepared by dissolving sucrose in the base juice, while the same methodology developed by Monteleone et al. [53] was used for preparation of the Cp variants.

Phenotypic responses to PROP (Part 4, Figure 1) were instead collected in duplicate from 10 mL aqueous solutions with 0.5447 g/L of PROP [53]. All stimuli were prepared the day before, stored at 4 °C overnight, and brought to ambient temperature 2 h prior to testing. Both food models and PROP solutions were served in 80 cc plastic glasses at ~20 °C, coded with a random 3-digit code, and tested under white warm light for liking and red light for intensity tasks (Figure 1).

#### 2.3.2. Training

Each sensory assessment was preceded by detailed instructions on the psychophysical scaling methods to be used for the upcoming tasks. Before the liking task (Part 2, Figure 1), participants were trained to use the Labeled Affective Magnitude scale (LAM, 0 = greatest imaginable dislike, 100 = greatest imaginable like) according to common practices [57]. Subsequently, before the intensity tasks outlined in Parts 3 and 4 (Figure 1), special attention was given to minimizing potential artifacts in the use of the gLMS (Table 2) by (a) clarifying the meaning of the scale anchors and the sensory attributes being evaluated, providing simple descriptions (e.g., a dry mouth feeling led by unripe fruits for astringency) along with relevant food examples (e.g., lemon juice for sourness), (b) encouraging the use of the full scale to avoid categorical behaviors and to distinguish the intensity of a stimulus from its hedonic value, and (c) guiding participants to base the ratings on their daily sensory experiences across various modalities [40,59,62].

For individual calibration, participants provided psychophysical responses to 10 extraoral stimuli shortly after rating the intensity of oral sensations from each variant of both Gj and Cp series (n = 4 × 2) or PROP solution (n = 2). The items were presented in a fixed order, and stimuli representing various theoretical ranges on the gLMS were included in each set of products. Collectively, systematic differences were observed in parallel with the expected magnitude of the orientation stimuli (Appendix A). Furthermore, the effectiveness of the gLMS training was supported by both girls and boys rating all extraoral stimuli as equally intense, indicating consistent use and interpretation of the scale (Appendix A).

#### 2.3.3. Sensory Assessments

In the hedonic task (Part 2, Figure 1), ratings were obtained from two independent sets of four variants each of Gj and Cp (Table 3) using the LAM scale (Table 2). Participants were instructed to hold the whole sample of Gj (20 mL) or a full spoon of Cp (20 g) in their mouth for 5 s before swallowing and then rate their liking.

The same quantity of sample was used for the intensity assessment (Part 3, Figure 1). Unlike the preceding task, participants were asked to keep each variant of Gj and Cp in their mouth for 7 s, then swallow and wait 5 s before evaluating the perceived intensities using the gLMS scale (Table 2). The Gj series was always presented first, followed by the Cp series after a 5 min break during which participants engaged in logic games to maintain motivation (Figure 1). In contrast, the variants were presented in different fixed orders across the liking and intensity tasks (Table 3). This approach was designed to minimize excessive fluctuations in perceived intensities that could lead to inflated responses, prevent participants from associating the same presentation order with both tasks, and induce similar perceptual biases across individuals, thereby facilitating subsequent comparisons of sensory responses [63]. For the same purpose, the sensory ballot was presented with a fixed sequence (Table 3): the target sensation (sweetness) was rated first, followed by relevant attributes to Gj (sour, bitter) or Cp (bitter, astringency), with flavor evaluated last [63]. Before sensory evaluations, participants were asked to declare any allergies and/or intolerances to Gj and Cp ingredients (Table 3).

For PROP phenotyping (Part 4, Figure 1), participants retained each solution in their mouth for 10 s, then expectorated and waited 20 s before reporting the perceived bitterness (gLMS, Table 2). The average PROP values were used after confirming similar ratings among replicates (W = 13,858, *p* = 0.295).

A 60 s break was enforced after each tasting, during which participants were provided with mineral water and plain crackers to cleanse their palate.

### 2.4. Questionnaires

#### 2.4.1. The Patient Health Questionnaire (PHQ-9)

To assess depressive symptoms, we utilized the Italian version of the 9-item Patient Health Questionnaire [46], which has demonstrated psychometric soundness with diverse Italian adolescent populations, e.g., [64,65]. The PHQ-9 is widely recognized as the gold standard for diagnosing depressive disorders, exhibiting high sensitivity (89%) and specificity (88%) in primary care settings [66]. A 4-point Likert scale (0 = never, 3 = nearly every day) is employed to quantify the frequency of depressive symptoms experienced over the past 2 weeks. The sum of the nine items yields a total score ranging from 0 to 27, with higher values reflecting greater severity of depressive symptoms. In this study, the PHQ-9 demonstrated good internal consistency (α = 0.818).

#### 2.4.2. The Screen for Child Anxiety Related Emotional Disorders (SCARED)

Anxiety levels were operationalized using a subset of the 38-item Screen for Child Anxiety Related Emotional Disorders adapted for the Italian adolescent population [60,67]. The SCARED employs a 3-point Likert scale (1 = almost never, 3 = often) to measure the frequency of anxiety-related symptoms across five factors: generalized anxiety (9 items), social anxiety (4 items), school anxiety (4 items), panic disorders (13 items), and separation anxiety (8 items). In the present study, only the generalized, school, and social anxiety subscales were collected. Each domain was scored by summing up the relevant items, with higher values indicating a greater inclination toward anxiety-related symptoms. The internal consistency of the subscales was acceptable to good, with Cronbach’s αs of 0.821, 0.747, and 0.684 for the generalized, social, and school anxiety, respectively.

#### 2.4.3. The Goldberg’s Big Five Questionnaire (BIG-5)

The facets of personality were assessed using the 30-item Goldberg’s Big Five questionnaire [68], which has been validated for use with Italian adolescents by Klimstra et al. [58]. Each personality dimension (agreeableness, conscientiousness, emotional stability, extraversion, and openness) was evaluated with six items on a 7-point Likert scale (1 = does not apply to me at all, 7 = applies to me very well). Composite subscale scores were then computed by averaging the ratings for each subscale, with the control items reversed. Each domain showed acceptable to good internal consistency, with Cronbach’s αs of 0.801 for agreeableness, 0.838 for conscientiousness, 0.778 for emotional stability, 0.845 for extraversion, and 0.671 for openness. For the purposes of this study, the emotional stability dimension was used and reversed to represent its opposite, neuroticism, which reflects the tendency to experience negative affective states [43].

#### 2.4.4. The Adult Picky Eating Questionnaire (APEQ)

As a food-related comorbidity of depressive symptoms [33], the Italian version of the 20-item Adult Picky Eating Questionnaire [55,69,70] was used to evaluate picky eating, defined as the reluctance to try familiar and novel foods [33]. Participants were asked to indicate how frequently they engage in picky eating behaviors on a 5-point Likert scale (1 = “Never”; 5 = “Always”) across four dimensions: meal presentation (7 items), food variety (4 items), meal disengagement (3 items), and taste aversion (6 items). Scores were calculated as the average of all 20 items or within each subscale, with higher values indicating greater pickiness.

Due to the absence of validated measures for assessing picky eating among Italian adolescents and our prior validation being specific to adults over 18 [55], only the APEQ global score was used in this study. To preliminarily assess its psychometric properties, we tested its internal consistency (α = 0.808), convergent and discriminant validity (Appendix A), and test-retest reliability, all yielding adequate results. For further details, please refer to Appendix B.

#### 2.4.5. The Body Image Dimensional Assessment (BIDA)

Lastly, participants completed the 4-item Body Image Dimensional Assessment [61], which measures the discrepancy between an individual’s current body image and their idealized physique. Participants viewed four anonymized body shapes representing increasing weights along with a line scale from 1.8 to 5.2, before rating how closely their actual body shape aligned with their ideal figure (body dissatisfaction), the most attractive shape for the opposite sex (sexual body dissatisfaction), and the appearance of same-sex and -age peers (compared body dissatisfaction). Three sub-scores (ranging from −100 to 100) can be computed using the formulas provided by Sánchez-Miguel et al. [61], and a total BIDA score was derived as the mean of their absolute values, with higher scores reflecting greater body dissatisfaction [61].

Although validated only for Spanish adolescents aged 12–15 [61], the BIDA was still preferred over the recently validated 8-item BIBA [71] in the Italian context, as the latter was designed for younger children (ages 6–13). After an initial psychometric check, the BIDA total score showed good internal consistency (α = 0.763), strong inter-item correlations (Appendix A), and evidence of convergent and discriminant validity (Appendix A). Further details are provided in Appendix C.

#### 2.4.6. Demographic and Lifestyle Variables

Alongside the remote (Table 1) and lab (Table 2) sessions, a range of demographic and lifestyle-related measures were collected. These included, but were not limited to, gender, age, and weight and height, with the latter used to calculate body mass index (BMI) in kg/m^2^. Additionally, physical activity levels were estimated using the Italian short form of the International Physical Activity Questionnaire (IPAQ) [51].

#### 2.4.7. The EPIC Food Frequency Questionnaire (EPIC-FFQ)

In the week preceding the lab session (Figure 1), participants filled out a paper-based version of the 163-item EPIC-FFQ, which was recently validated for Italian adolescents [45], to monitor their habitual diet over the past year. Participants were asked to report consumption frequencies for both individual foods (e.g., vegetables) and recipes (e.g., pasta, pizza), along with habitual cooking methods (e.g., baked, boiled, fried, grilled) across daily, weekly, monthly, and yearly intervals. To ensure accuracy, the questionnaire included images depicting portion sizes (small, medium, large) for various recipes (e.g., soups, stew) or food items within each food group (e.g., carrots for vegetables, cod for fish). Dietary data were treated according to Pala et al. [72] to estimate daily intake of energy (Kcal) and a list of macro- (e.g., carbohydrates, fats, proteins) and micronutrients (e.g., B vitamins, minerals) before further processing (Section 2.5).

### 2.5. Data Analysis

Given that the majority of variables did not meet the normality assumptions, gender differences in severity of depressive symptoms and their psychosocial correlates (generalized anxiety, neuroticism, picky eating, and body dissatisfaction) were first assessed using the Wilcoxon Rank Sum Test (W). To prepare for further analyses, the PHQ-9 scores, which exhibited right skewness (γ = 1.421), were transformed using the Yeo–Johnson method to approximate normality (γ = 0.030). Additionally, psychophysical responses to oral sensations evoked by each variant of Gj (sweet, sour, bitter) and Cp (sweet, bitter, astringent) were individually summed to derive six scores of global responsiveness (Ʃ), as outlined by Piochi et al. [73]. Importantly, hedonic and intensity ratings were available for 231 and 227 individuals, as one participant reported constraints to ingredients in Gj, and five in Cp.

Separate general linear models (GLMs) were then fitted to evaluate whether the associations between sensory perception and depressive symptoms differed by gender. In these models, the PHQ-9 scores served as the dependent variable, while both the main effects and the interactions between gender and each of the six taste global scores were included as predictors. To control for possible psychosocial confounders, each GLM was re-run with comorbid states or traits of depression included as covariates. According to the results, the global taste scores that exhibited significant interactions with gender as predictors of depressive symptoms, either with or without adjustments, were retained for further analysis. Data were then stratified by gender, and subsequent analyses were conducted separately for girls and boys.

Next, partial Spearman’s rank correlation coefficients (ρ) were computed to examine gender-specific relationships between depressive symptoms and dietary habits, adjusting for age, BMI, and physical activity. Before analysis, nutrient data were screened for misreporting and implausible extreme values, as recommended by Welch et al. [74]. In brief, participants with more than 10 missing items (n = 8) due to misreporting or duplicate responses were excluded [74]. Next, basal metabolic rates specific to gender and age were estimated for each participant using the Schofield equation [75], and outliers (n = 6) were identified as those whose ratio of habitual energy intake to basal metabolic rate fell within the top or bottom 1% of the distribution [76]. As a result, data from 218 participants who had a demographic and lifestyle background similar to that of the original population (Appendix A), were retained for analysis. Nutrient data were then individually adjusted for daily energy intake (Kcal) using the residual method [77] to account for variations in energy needs and to prevent inflation of the results.

Lastly, moderation analysis was applied to test whether varying vulnerabilities to correlates of depression led to variations in oral acuity that may prompt undesired dietary choices with long-term effects on mood. Moderation analysis evaluates how the effect of an independent variable (oral acuity) on a dependent variable (nutrient intake) varies with the level of a moderating variable (correlates of depression) [78]. Accordingly, each energy-adjusted nutrient was treated as a dependent variable, with both main effects and interaction terms between each global taste score (independent variable) and each comorbidity of depression (moderator) included as predictors. Again, these analyses were adjusted for age, BMI, and physical activity.

Bootstrapping with 10,000 iterations was used to robustly estimate 95% confidence intervals for all models, and continuous variables were scaled to unit variance prior to analysis to facilitate interpretability. Effect sizes are presented as bootstrapped standardized β coefficients, with squared semi-partial correlations (sr^2^) included in moderation analysis to quantify the unique variance explained by each predictor [78]. The absence of severe multicollinearity and autocorrelation was confirmed by Variance Inflation Factor (VIF) values < 10 [79] and non-significant results from the Durbin–Watson test (*p* > 0.05), respectively. Lastly, data are summarized as median ± interquartile range (IQR) where applicable, with all tests being two-tailed and statistical significance set at *p* < 0.05. Data analysis was conducted using R version 4.3.1 [80].

## 3. Results

### 3.1. Rates of Depression and Gender Differences in Depressive Symptoms and Comorbid Traits

According to standard clinical cut-offs [46], our sample overall showed mild (PHQ-9 = 5–9) depressive symptoms (median ± IQR = 7 ± 5). Nonetheless, at least moderate depressive symptoms (PHQ-9 ≥ 10) were observed in 21.6% of participants. In detail, 1.7%, 27.6%, 49.1%, 13.4%, 6.0%, and 2.2% of adolescents reported experiencing no (PHQ-9 = 0), minimal (PHQ-9 = 1–4), mild (PHQ-9 = 5–9), moderate (PHQ-9 = 10–14), moderately severe (PHQ-9 = 15–19), and severe (PHQ-9 ≥ 20) depressive symptoms over the past two weeks, respectively. Notably, 36.1% of adolescent girls exhibited at least moderate depressive symptoms compared to 11.1% of boys.

As a result, significant (*p* < 0.001) gender differences in depression and related comorbidities emerged (Figure 2). Girls reported greater depressive symptoms, a higher inclination toward generalized anxiety, increased levels of neuroticism, and more engagement in picky eating behaviors compared to boys. In contrast, no significant differences were found between genders regarding overall body dissatisfaction (W = 6532.5, *p* = 0.977).

### 3.2. Associations Between Sensory Perception and Depression by Gender

Next, we examined whether the associations between PHQ-9 scores and the six global scores of oral acuity varied by gender. Across all models, gender consistently showed a significant main effect on depressive symptoms (*p* < 0.001), whereas the opposite was true for all global taste scores (Table 4). This suggests that gender was a more influential predictor of PHQ-9 scores than orosensory responsiveness.

However, the relationship between sensory perception and depression was gender-specific. Notably, significant interaction effects were found in girls for global acuity for bitterness in Gj and astringency in Cp. In contrast, no significant or almost significant interactions were observed for sweetness in either food model, sourness in Gj (*p* = 0.055), or bitterness in Cp (Table 4). These findings indicate that greater responsiveness to alarming oral sensations predicted higher depressive symptoms in girls, but not in boys.

Similar patterns emerged after adjusting for psychosocial correlates of depression (Table 5). The main effects of global taste scores remained non-significant, while the predictive value of gender lost significance after adjustment. As expected, generalized anxiety and neuroticism were the strongest predictors of PHQ-9 scores across all models, with higher internalizing symptoms systematically linked to higher levels of depression.

Consistent with our previous findings, no significant interactions between gender and sweetness were observed in either food model. Similarly, the positive association between acuity for alarming oral sensations and PHQ-9 scores remained exclusive to girls, with significant interaction effects found for bitterness in Gj and for both bitterness and astringency in Cp (Table 5). Notably, both the crude (Appendix A) and adjusted (Appendix A) models showed no evidence of severe multicollinearity or autocorrelation (Durbin–Watson tests, *p* > 0.05), with VIFs well below the commonly used threshold of 10 [79]. Based on these results, only the global scores related to alarming oral sensations were retained for further analysis.

### 3.3. Associations Between Dietary Habits and Depressive Symptoms by Gender

After stratifying by gender, we examined the correlations between depressive symptoms and dietary habits. Several commonalities emerged between girls and boys, though depressive symptoms were associated with a greater number of dietary outcomes in girls (Table 6).

In both genders, PHQ-9 scores were positively correlated (*p* < 0.05) with alcohol intake and exhibited inverse associations with daily consumption of proteins, vegetable fats, oleic acid, and vitamin E (Table 6).

It is noteworthy that, in addition to a positive correlation with carbohydrate intake, depressive symptoms were inversely correlated (*p* < 0.05) with the intake of a lengthy list of beneficial nutrients exclusively in girls. These included, but were not limited to, fibers, polyunsaturated fats, vitamin B6, vitamin D, and zinc. In contrast, habitual intake of vegetable proteins showed an inverse correlation with PHQ-9 scores in boys, but not in girls (Table 6).

###  3.4. Moderation Analysis

Lastly, we performed several moderation analyses to examine whether varying vulnerabilities to comorbidities of depression influenced the link between global responsiveness to alarming oral sensations and diet. In general, psychosocial correlates of depression primarily moderated the associations between dietary habits and responsiveness to alarming oral sensations in girls (Figure 3). For girls, the moderation effect from all significant models (F *p*-value < 0.05) accounted for additional variance (sr^2^) ranging from 1.8% to 7.8% for generalized anxiety, 0.0% to 7.7% for neuroticism, 1.0% to 6.8% for picky eating, and 4.7% to 9.8% for body dissatisfaction. In contrast, for boys, the incremental variance attributed to these variables was generally lower and often non-significant (*p* > 0.05), ranging from 0.0% to 5.5% for generalized anxiety, 0.5% to 4.1% for neuroticism, 0.0% to 3.1% for picky eating, and 0.0% to 0.9% for body dissatisfaction.

Specifically, comorbidities of depression moderated (*p* < 0.05) the association between responsiveness to bitterness, astringency, and (to a lesser extent) sourness and the consumption of various healthful nutrients (Figure 3). For instance, generalized anxiety (sr^2^ = 0.078), neuroticism (sr^2^ = 0.057), and picky eating (sr^2^ = 0.045) moderated the negative association between bitter taste acuity and habitual fiber intake. Similarly, neuroticism exerted a moderating role on the inverse association between bitterness perception and habitual intake of proteins (sr^2^ = 0.040), calcium (sr^2^ = 0.077), and phosphorus (sr^2^ = 0.040). Additionally, body dissatisfaction was found to moderate the negative association between the intake of vitamins B1 (sr^2^ = 0.050), B2 (sr^2^ = 0.050), B3 (sr^2^ = 0.062), and B6 (sr^2^ = 0.092) and responsiveness to bitterness (in Cp), astringency, bitterness (in Gj), and sourness, respectively (Figure 3). In other words, adolescent girls with higher levels of comorbid traits of depression showed stronger negative associations between responsiveness to alarming oral sensations and the consumption of these nutrients.

Importantly, all models were free from multicollinearity (VIF < 10) [79] and autocorrelation issues, with VIFs ranging from 1.005 to 1.837 for girls and 1.014 to 1.259 for boys. Additionally, the results from the Durbin–Watson test were non-significant (*p* > 0.05) across all analyses.

To facilitate comprehension of the moderating effects, Figure 4 depicts examples of how varying levels of correlates of depression influence the strength of associations between sensory perception and dietary intake by gender. Full details on bootstrapped standardized β coefficients and 95% confidence intervals, as well as *p*-values, sr^2^, and adjusted R^2^ for all significant models (F *p*-value < 0.05) highlighting a moderating effect of generalized anxiety, neuroticism, picky eating, and body dissatisfaction on the relationship between acuity for alarming oral sensations and nutrient intake by gender are provided in Appendix A.

## 4. Discussion

In this study, we investigated how gender influences the associations between sensory perception, depression, and diet during adolescence. Given the differing vulnerabilities between girls and boys, e.g., [7], we further examined whether common comorbidities of depression moderated the impact of sensory perception on dietary choices. Our findings first confirmed that girls are more susceptible to mood disturbances and revealed that acuity for bitterness and astringency is positively associated with depressive symptoms exclusively in this group. Additionally, we corroborated known associations between depression and poor dietary habits and found that higher levels of internalizing symptoms in girls exacerbated the inverse relationship between heightened responsiveness to alarming oral sensations and the intake of several nutrients.

### 4.1. Adolescent Girls Exhibit Higher Levels of Internalizing Symptoms than Boys

The concerning global trends in the prevalence of depression among adolescents, e.g., [3,7], were also evident in our sample. Using a PHQ-9 score ≥ 10 [46] as a sensitive (85%) and specific (89%) indicator of early major depressive disorders [66], 21.6% of adolescents exceeded this threshold. This finding is consistent with recent data from similar regions, where 16% of European [3] and 20.8% of Italian adolescents [81] are currently at risk of developing clinical depression. Moreover, we corroborated established gender differences in susceptibility to internalizing symptoms. In our sample, girls reported depressive symptoms at a rate three times higher than boys (36.1% vs. 11.1%), thus slightly surpassing the 2017–2018 estimates (28.7% vs. 10.4%) from a larger Italian sample [81] and underscoring the growing incidence of mental health difficulties among adolescents [3,7].

It was therefore unsurprising to observe girls exhibiting higher levels of common comorbid traits of depression, such as generalized anxiety, neuroticism, and picky eating [7,33,42,43]. However, interpreting the latter finding is less straightforward. The literature on variations in the engagement of picky eating behaviors by gender is sparse and inconsistent, often focusing on younger children and varying in assessment tools and definitions of picky eating itself [82]. For instance, Marchi and Cohen [83] found that pickiness was more common in girls (n = 326) than in boys (n = 333) across the ages of 1–10, 9–18, and 11–21 over a 10-year follow-up. In contrast, no significant gender differences were found in other cross-sectional [84] or prospective [85] studies.

Despite this mixed evidence, it is reasonable to speculate that our findings reflect underlying gender differences in domains of anxiety, which have been reported to prompt selective eating behaviors, e.g., [33]. This is further supported by systematic gender disparities observed in the links between pickiness and domains of anxiety within our sample, with stronger associations in girls for generalized (girls: ρ = 0.334, *p* < 0.001; boys: ρ = 0.205, *p* = 0.017), social (girls: ρ = 0.313, *p* = 0.002; boys: ρ = 0.193, *p* = 0.025), and school anxiety (girls: ρ = 0.371, *p* < 0.001; boys: ρ = 0.205, *p* = 0.017). However, further research is needed to confirm or refute this hypothesis within diverse nonclinical adolescent populations.

Surprisingly, our data also revealed no gender variations in overall body dissatisfaction. This may tentatively be attributed to the multifaceted nature of the BIDA total score, which assesses various aspects of body dissatisfaction. Indeed, girls felt their actual body shape was more rounded compared to their ideal figure, whereas boys perceived their physique to be leaner than what they viewed as most attractive to the opposite sex (Appendix A). Hence, while our findings contrast with those of Sánchez-Miguel et al. [61], they align with the current understanding that weight concerns and the pursuit of muscularity are key contributors of body dissatisfaction for girls and boys, respectively (for review, see [86]).

### 4.2. Greater Responsiveness to Bitterness and Astringency Is Associated with Higher Severity of Depressive Symptoms in Adolescent Girls, but Not in Boys

We found no main effects of sensory responsiveness on depressive symptoms, thus suggesting that taste *per se* is not a relevant predictor of depressive symptoms in healthy adolescents. While comparisons with the existing literature should be made cautiously due to differing methods for assessing taste function that may not be correlated [40], our result aligns with prior research involving nonclinical adults showing no associations between depression and thresholds for sweet, sour, bitter, or salty tastes [35], intensity ratings of chocolate and vanilla milks [35], or recalled saltiness intensities for 10 food items [36]. However, it contrasts with studies observing either positive [31,34] or negative [37] associations between depressive symptoms and intensity ratings for sweet and bitter-tasting aqueous solutions.

Nonetheless, we provide evidence that sensory perception may differentially predict mood disturbances by gender. Specifically, responsiveness to bitterness and astringency was associated with the severity of depressive symptoms exclusively in girls, an effect observed in both the crude and adjusted models. This finding is somewhat consistent with a Japanese work involving young adults (n = 70, mean age ± SD = 20.7 ± 0.3 y) that explored gender- and menstrual-cycle-related associations between internalizing symptoms (depression, anxiety) and sweet taste recognition thresholds [87]. In that study, depression was positively correlated with sweet taste sensitivity (i.e., decreased thresholds) uniquely in women during the luteal phase, with no correlations detected in men or in women during the follicular phase.

However, a similarity with the findings of Nagai et al. [87] exists in the gender-specific association between a high responsiveness to oral stimuli and the severity of internalizing symptoms. Intriguingly, the luteal phase group was the sole group to show a positive correlation between sweet sensitivity and anxiety, which was independent of baseline differences in anxiety between genders or menstrual cycle groups [87]. While the authors hypothesized that this pattern might be attributed to higher levels of luteinizing hormones [87], we can further speculate that the predictive value of sensory perception regarding depressive symptoms might become more evident when comparing individuals experiencing distinct physiological states and/or possessing varying vulnerabilities to psychological constructs that predispose them to heightened affective responses [88,89].

Although the menstrual cycle was not considered in our study, we still lend support to this notion through three key points. First, adolescent girls exhibited higher anxiety-related traits relative to boys. Second, putative physiological influences on findings can be reasonably ruled out, as no gender differences were observed in either PROP acuity (Appendix A) or global responsiveness to oral sensations (Appendix A). Lastly, external cues are unlikely to fully account for the observed differences, as both genders had similar demographic and lifestyle backgrounds (Appendix A), were equally familiar (Food Familiarity Questionnaire, Figure 1) with grapefruit juice (W = 6158, *p* = 0.423) and dark chocolate (W = 6051, *p* = 0.294), and reported similar hedonic ratings for all variants of Gj and Cp, except for the highest sucrose concentration in both food models (Appendix A).

### 4.3. Comorbidities of Depression Moderate the Links Between Sensory Perception and Dietary Habits with Long-Term Adverse Effects on Mood in Adolescent Girls

Consistent with the existing literature, e.g., [10,11,12,13], we noted that various dietary outcomes were inversely correlated with PHQ-9 scores, with this relationship being more pronounced in girls. Specifically, alcohol use combined with low intake of proteins, vegetable fats, oleic acid, and vitamin E was associated with depressive symptoms in both genders. Similar patterns, though not statistically significant in boys, were observed with low daily consumption of animal proteins, mono- and polyunsaturated fats, linoleic acid, B vitamins (B1, B6, B9), iron, phosphorus, and zinc.

Several mechanisms have been proposed to explain how inadequate intake of these nutrients might exacerbate depression, and vice versa. Depression may prompt unhealthy eating habits, and nutrient deficiencies may impair brain function by disrupting the synthesis and activity of monoamines (serotonin, norepinephrine, dopamine) or by increasing oxidative stress, both of which are thought to be involved in the pathophysiology of depression (for review, see [14]). For example, amino acids from both animal and plant proteins are key for monoamine production, while omega-3 fatty acids (e.g., linolenic acid) enhance serotonin function and are believed to counteract oxidative stress through their potent anti-inflammatory effects. Similarly, B-group vitamins and iron support neurotransmitter metabolism, and zinc is essential for the optimal functioning of brain regions involved in mood regulation and cognitive functions [10]. Consequently, it is not surprising that dietary supplementation with nutrients involved in the homeostasis of brain functions has frequently led to significant improvements in depressive symptoms across diverse adolescent populations [10]. Our findings thus reinforce the substantial evidence supporting a mutual link between adequate nutrition and mental health during adolescence.

In this context, we observed that comorbidities of depression exacerbated the negative relationships between acuity for alarming oral stimuli and the intake of various nutrients associated with higher PHQ-9 scores, particularly among girls. Importantly, the overall direction of these effects was consistent across genders and aligned with current knowledge, e.g., [15,90], with greater responsiveness generally serving as a barrier to consuming beneficial nutrients that also support mood regulation (Appendix A). However, in models accounting for the moderating effects of correlates of depression, significant associations emerged almost exclusively in girls. This suggests that other biological, non-biological, or psychosocial variables may better explain how variations in perception of alarming oral sensations affect dietary habits in adolescent boys, warranting further investigation.

Conversely, we can tentatively discuss why depression-related states or traits might strengthen the association between sensory perception and suboptimal dietary choices in girls. Alarming oral sensations, which signal potentially aversive post-ingestive effects, naturally elicit negative arousal. Even minor stimulation can trigger a cascade of psychobiological responses, which individuals may cope with differently before deciding whether to accept or reject the stimulus. Beyond intensity, the complexity and novelty of sensory stimuli also contribute to arousal, with responses shaped by expectations about the upcoming experience. A discrepancy between expected and perceived sensations may therefore lead to exaggerated attention on specific product cues, thereby intensifying negative emotional states [39]. This might explain why the associations between bitterness in Cp and diet were more pronounced than those observed in Gj. It is possible that participants had well-formed expectations about the taste qualities of both food models, with bitterness being more anticipated in Gj than in Cp. Therefore, a mismatch between expected and actual taste might have evoked negative arousal, which was possibly further promoted by participants’ greater familiarity with chocolate puddings (W = 838, *p* < 0.001). This, in turn, would have magnified the discrepancy between anxiety-related traits and highlighted stronger moderating effects. However, further studies are needed to confirm this hypothesis.

Alternatively, it is noteworthy that a higher vulnerability to internalizing symptoms often reflects a generalized hyperresponsiveness to environmental inputs and stressors, such as noise, tactile stimuli [91], pain [92] or danger-signaling smells [93]. While this trend is observed in both girls and boys, notable gender differences in coping strategies for stress exist [89]. Generally speaking, girls are more likely to notice and report even subtle emotional changes, while boys tend to exhibit more passive behaviors that shift toward a more feminine coping style only once signs of distress become evident [89,94]. Such heightened reactivity to stressors might thus translate into intensified attentional bias and negative affective states in response to negatively arousing stimuli, thereby reinforcing avoidant behaviors when alarming oral sensations are experienced in food. Conversely, a similar pattern might only emerge in boys when depression-related states or traits and extremely intense arousing sensations occur concurrently. To either confirm or refute this hypothesis, future studies with adolescent populations exhibiting similar levels of psychosocial correlates of depression among genders and exposed to food stimuli that combine multiple sources of arousal are necessary.

### 4.4. Strengths and Limitations

To our knowledge, this study is the first to examine the associations between depressive symptoms, taste perception, and dietary habits in a nonclinical adolescent population. Its key strengths include the use of one of the largest samples ever employed in this line of research, a thorough data collection protocol, and the use of real food models to assess a range of oral sensations at intensities that closely mirror real-life experiences. In response to calls for novel research on the role of taste in adolescent food choices [24], we also extend the existing literature by testing previously overlooked oral sensations (sourness, astringency). Furthermore, this work presents the first empirical evidence linking astringency perception with depressive symptoms in healthy adolescent girls. Lastly, we provide preliminary support for the psychometric validity of the APEQ and the BIDA for use with Italian adolescents, which have the potential to offer deeper insights into the determinants of eating habits in this age group, pending their comprehensive psychometric validation.

Nevertheless, our results should also be interpreted in light of several limitations. Firstly, while we employed ecologically valid food stimuli and sensory assessments, we did not test other relevant tastes (salty, umami) or chemesthetic sensations (e.g., pungency) that may influence adolescents’ dietary habits. Additionally, our focus was on a limited set of food models, which elicited a narrow range of sensory qualities typically encountered in daily experiences. Whether the results can be replicated with a broader variety of foods differing in sensory, chemical, and physical properties should thus be probed in future investigations.

Secondly, the dietary data were based on self-reports, which can be subject to motivational and recall biases, as well as under- or over-reporting practices [95]. Despite meticulous data preprocessing to minimize these biases and a strong alignment with existing knowledge, the potential for inaccuracies remains. In order to enhance the reliability of data, future research should incorporate more precise and less burdensome approaches, such as technological aids that facilitate memory and portion estimation, ideally within repeated measures frameworks. In this vein, the integration of 24-hour recalls with digital tools (e.g., mobile apps, image-based assessments) holds promise in mitigating prevalent sources of error in adolescent populations.

Thirdly, the gender-based sensitivity analysis may have lacked sufficient statistical power to detect subtle yet meaningful effects, particularly given the relatively modest variance explained by the moderation models (Appendix A). Besides suggesting that other factors, including genetic, non-genetic (e.g., oral microbiome), and psychosocial variables not included in this study, might further elucidate how taste perception influences dietary outcomes and potentially affects mood in the long term, this prompts the need to devote future efforts to include larger and adequately powered samples.

Fourthly, the internalizing symptoms investigated herein are inherently interrelated and might have acted as confounding or overlapping variables in the observed associations. Case–control studies or novel research involving non-clinically gender-stratified groups that differ in key psychosocial traits will be instrumental in further validating and extending our findings. Lastly, due to the study’s design, causality can not be inferred. Novel longitudinal studies are thus needed to determine whether taste perception indirectly exacerbates depressive symptoms through inadequate food choices or whether the opposite is true.

### 4.5. Conclusions

In addition to confirming a greater susceptibility to mood disturbances in girls compared to boys, along with associations between internalizing symptoms and habitual nutrient intake, this study methodologically contributes to expanding knowledge on picky eating and body image in Italian adolescents by supporting the initial psychometric validity of tools currently used across different age groups (APEQ) or geographic contexts (BIDA). More importantly, we present the first evidence linking heightened responsiveness to bitterness and astringency, evoked by variants of food models designed to mimic a range of intensities encountered in daily food experiences, with greater severity of depressive symptoms exclusively in girls. While our findings require replication across a broader range of food stimuli, oral sensations, adolescent populations, and longitudinal frameworks, they remain highly relevant in supporting future gender-tailored, sensory-based strategies aimed at promoting adherence to dietary patterns that may confer long-term benefits for mental health. In conclusion, this study advances our understanding of gender-specific vulnerabilities to mood disturbances during adolescence and suggests that the links between depressive symptoms, taste perception, and dietary habits are gender-specific, possibly shaped by differing coping strategies for internalizing symptoms.

## Figures and Tables

**Figure 1 nutrients-17-01653-f001:**
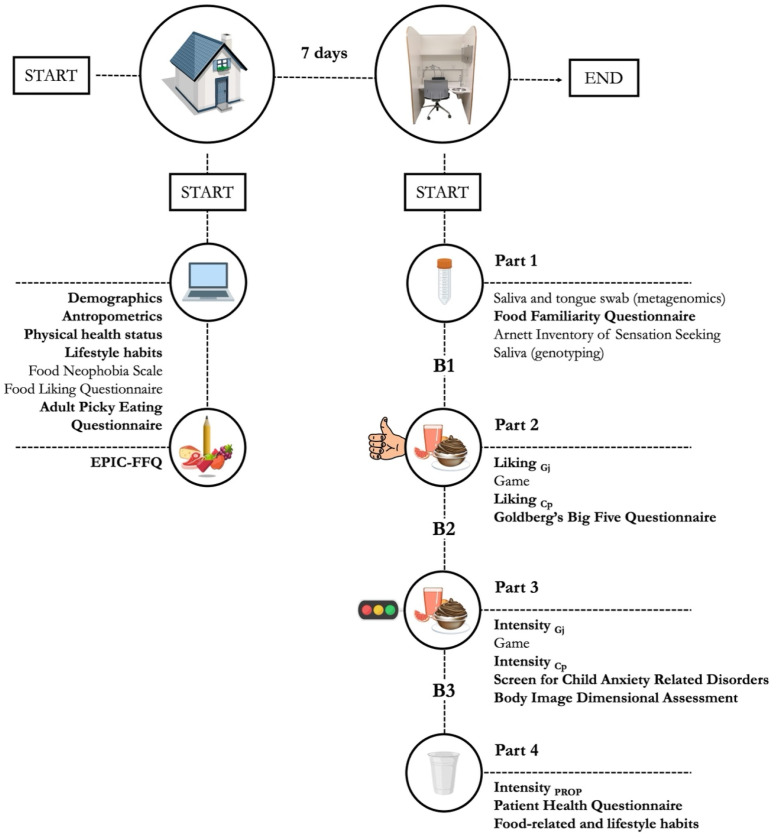
Graphical overview of data collection, with measures employed in this study highlighted in bold. Gj: grapefruit juice series; Cp: dark chocolate pudding series.

**Figure 2 nutrients-17-01653-f002:**
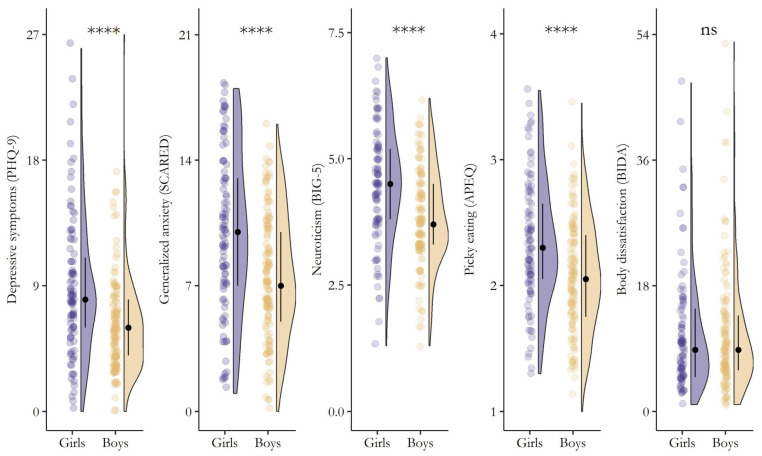
Differences (Wilcoxon Rank Sum Test) in the severity of depressive symptoms, generalized anxiety, neuroticism, picky eating behaviors, and body dissatisfaction between girls (slate blue) and boys (light tan). The plot shows raw data points (the “rain”), the kernel density estimate (the “cloud”), and the median (black filled circle) ± IQR (perpendicular black line). **** = *p* < 0.0001, ns = *p* > 0.05.

**Figure 3 nutrients-17-01653-f003:**
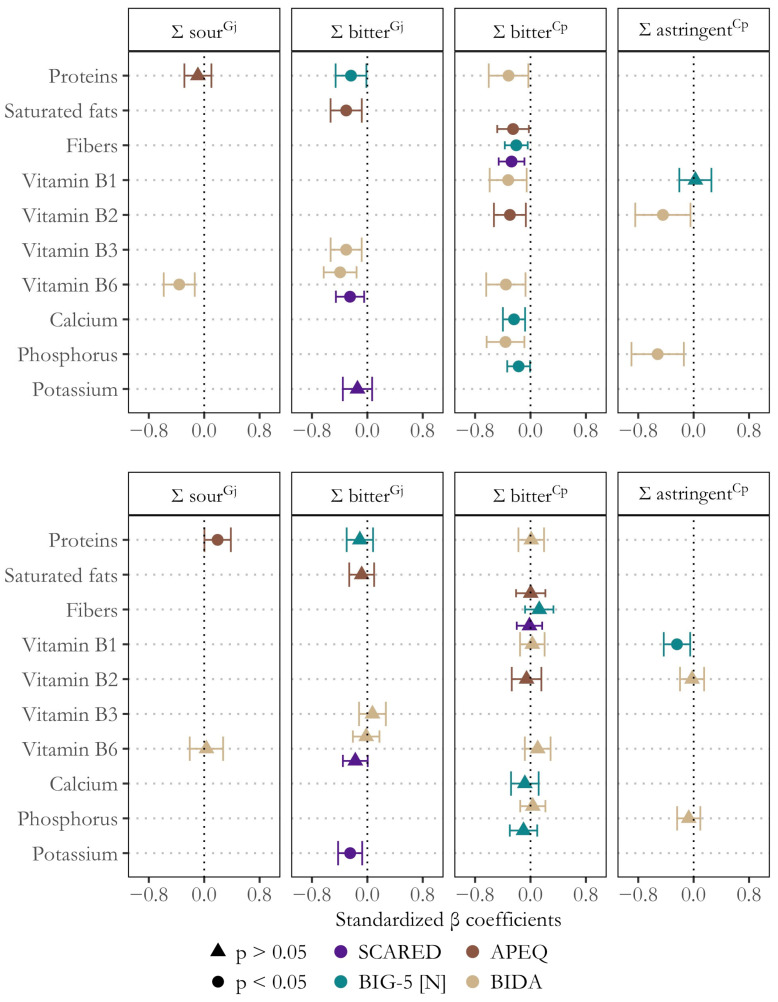
Significant moderating effects of depression comorbidities on the links between global responsiveness (Σ) to alarming oral sensations and nutrient intake in girls (**top**) and boys (**bottom**). The plot shows bootstrapped standardized β coefficients and 95% confidence intervals, along with *p*-values (circles: *p* < 0.05, triangles: *p* > 0.05). SCARED (generalized anxiety), BIG-5 [N] (neuroticism), APEQ (picky eating), and BIDA (body dissatisfaction).

**Figure 4 nutrients-17-01653-f004:**
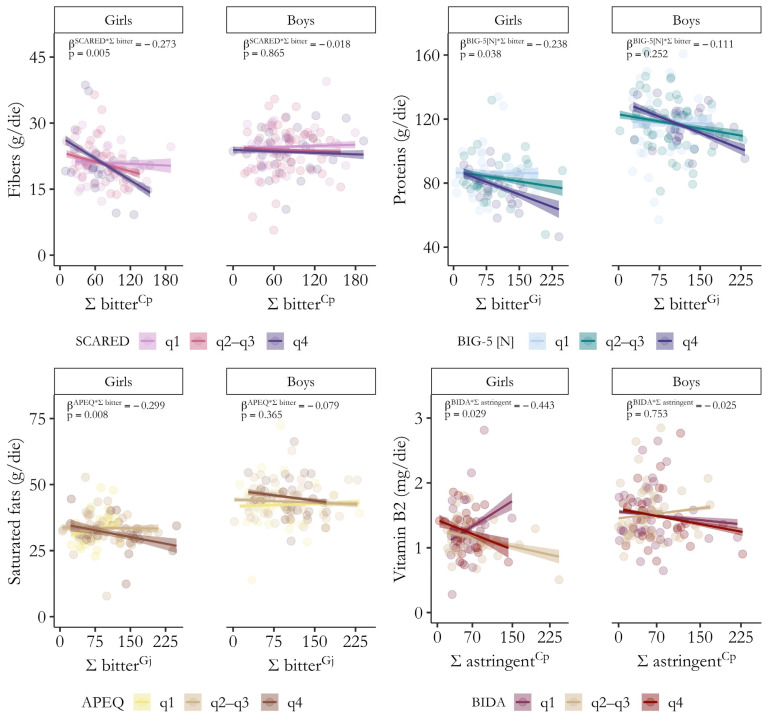
Significant moderating effects of depression comorbidities on the relationship between global responsiveness (Ʃ) to alarming oral sensations and dietary habits by gender. Simple slopes illustrate how internalizing symptom severity (q1: lowest quartile, q2–q3: second and third quartiles, q4: highest quartile) may exacerbate the association between sensory perception and nutrient intake (fibers: top left, proteins: top right, saturated fats: bottom left, vitamin B2: bottom right). Bootstrapped standardized β coefficients and *p*-values for the interaction term (Ʃ taste * correlates of depression) from moderation analyses are shown. SCARED (generalized anxiety), BIG-5 [N] (neuroticism), APEQ (picky eating), and BIDA (body dissatisfaction).

**Table 3 nutrients-17-01653-t003:** Food models and relative variants used in the present study. Ingredients (Brand), order of presentation of each variant within the liking and intensity tasks, sucrose concentrations added to each variant of Gj and Cp (g/kg), and the sensory ballot evaluated are listed. Target sensation is highlighted in bold.

Product	Ingredients (Brand)	Variant	Sucrose (g/kg)	Order (Liking)	Order (Intensity)	Oral Sensations
Grapefruit juice (Gj)	Sucrose (Zucchero, Eridania S.p.A., Genoa, Italy)	P01	0	2	4	**Sweet**, Sour, Bitter, Grapefruit
		P02	40	3	1	
	Grapefruit juice 100% (Puertosol, Eurospin Italia S.p.A, San Martino Buon Albergo, Italy)	P03	92	1	3	
		P04	160	4	2	
Chocolate pudding (Cp)	Sucrose (Zucchero, Eridania S.p.A, Italy)	P01	0	2	4	**Sweet**, Bitter, Astringent, Chocolate
	Chocolate pudding mix (Budino da zuccherare, Cameo S.p.A, Desenzano del Garda, Italy)	P02	60	3	1	
	Cocoa powder (Cacao Amaro Perugina, Nestlé, Assago, Italy)	P03	138	1	3	
	Water	P04	239	4	2	

**Table 4 nutrients-17-01653-t004:** Associations between global sensory responsiveness (Ʃ) and depressive symptoms as a function of gender. Bootstrapped β estimates, 95% confidence intervals, and *p*-values are provided. Statistically significant main and interaction effects are highlighted in bold. Gender [G]: girls, Gj: grapefruit juice, Cp: dark chocolate pudding.

Gj				Cp			
Predictors	β	95% CI	*p* Value	Predictors	β	95% CI	*p* Value
Ʃ sweet	0.089	−0.044–0.246	0.210	Ʃ sweet	0.089	−0.088–0.263	0.298
Gender [G]	0.526	0.268–0.792	**<0.001**	Gender [G]	0.548	0.296–0.795	**<0.001**
Ʃ sweet × Gender [G]	−0.161	−0.453–0.156	0.310	Ʃ sweet × Gender [G]	0.021	−0.248–0.270	0.897
Ʃ sour	−0.024	−0.126–0.093	0.670	Ʃ bitter	−0.027	−0.148–0.130	0.718
Gender [G]	0.538	0.288–0.786	**<0.001**	Gender [G]	0.550	0.287–0.808	**<0.001**
Ʃ sour × Gender [G]	0.260	−0.005–0.519	0.055	Ʃ bitter × Gender [G]	0.221	−0.104–0.555	0.186
Ʃ bitter	−0.031	−0.168–0.124	0.671	Ʃ astringent	0.036	−0.07–0.153	0.506
Gender [G]	0.522	0.278–0.766	**<0.001**	Gender [G]	0.528	0.282–0.761	**<0.001**
Ʃ bitter × Gender [G]	0.260	0.000–0.503	**0.050**	Ʃ astringent × Gender [G]	0.310	0.004–0.520	**0.048**
n	231			n	227		

**Table 5 nutrients-17-01653-t005:** Associations between global responsiveness (Ʃ) to oral sensations and depressive symptoms by gender, controlling for psychosocial comorbidities of depression. Bootstrapped standardized β coefficients and 95% confidence intervals, as well as *p*-values, are listed. Statistically significant main and interaction effects are highlighted in bold. Gender [G]: girls, Gj: grapefruit juice Cp: dark chocolate pudding, SCARED: generalized anxiety, BIG-5 [N]: neuroticism, APEQ: picky eating, BIDA: body dissatisfaction.

Gj				Cp			
Predictors	β	95% CI	*p* Value	Predictors	β	95% CI	*p* Value
Ʃ sweet	0.048	−0.084–0.205	0.499	Ʃ sweet	0.044	−0.108–0.203	0.546
Gender [G]	0.172	−0.049–0.409	0.125	Gender [G]	0.183	−0.051–0.417	0.117
SCARED	0.312	0.162–0.465	**<0.001**	SCARED	0.312	0.154–0.466	**<0.001**
BIG-5 [N]	0.204	0.045–0.350	**0.012**	BIG-5 [N]	0.206	0.050–0.356	**0.007**
APEQ	0.089	−0.035–0.209	0.145	APEQ	0.079	−0.042–0.201	0.216
BIDA	0.030	−0.077–0.123	0.576	BIDA	0.029	−0.083–0.127	0.631
Ʃ sweet × Gender [G]	−0.137	−0.375–0.092	0.229	Ʃ sweet × Gender [G]	−0.008	−0.242–0.203	0.920
Ʃ sour	−0.028	−0.133–0.101	0.645	Ʃ bitter	−0.063	−0.171–0.077	0.353
Gender [G]	0.193	−0.031–0.427	0.091	Gender [G]	0.189	−0.045–0.433	0.114
SCARED	0.299	0.148–0.452	**<0.001**	SCARED	0.315	0.164–0.466	**<0.001**
BIG-5 [N]	0.210	0.057–0.354	**0.008**	BIG-5 [N]	0.205	0.050–0.349	**0.007**
APEQ	0.082	−0.040–0.204	0.178	APEQ	0.079	−0.037–0.205	0.189
BIDA	0.034	−0.067–0.125	0.519	BIDA	0.050	−0.067–0.139	0.440
Ʃ sour × Gender [G]	0.155	−0.069–0.367	0.172	Ʃ bitter × Gender [G]	0.244	0.004–0.483	**0.047**
Ʃ bitter	−0.079	−0.202–0.059	0.242	Ʃ astringent	−0.040	−0.152–0.084	0.511
Gender [G]	0.186	−0.034–0.418	0.094	Gender [G]	0.186	−0.045–0.413	0.110
SCARED	0.294	0.145–0.445	**0.001**	SCARED	0.306	0.155–0.459	**<0.001**
BIG-5 [N]	0.221	0.069–0.362	**0.005**	BIG-5 [N]	0.193	0.036–0.343	**0.013**
APEQ	0.074	−0.046–0.190	0.222	APEQ	0.079	−0.043–0.202	0.209
BIDA	0.044	−0.054–0.133	0.382	BIDA	0.047	−0.063–0.143	0.403
Ʃ bitter × Gender [G]	0.238	0.019–0.442	**0.035**	Σ astringent × Gender [G]	0.269	0.034–0.458	**0.029**
n	231			n	227		

**Table 6 nutrients-17-01653-t006:** Partial correlation between the severity of depressive symptoms and nutrient intake by gender. Spearman’s ρ coefficients, adjusted for age, BMI, and physical activity, along with *p*-values, are listed. Statistically significant values (*p* < 0.05) are indicated in bold.

Nutrients		Girls		Boys	
		ρ	*p* Value	ρ	*p* Value
Carbohydrates	Carbohydrates	0.171	**0.004**	0.053	0.303
	Simple sugars	0.041	0.498	0.060	0.245
	Fibers	−0.161	**0.007**	−0.049	0.344
Fats	Fats	−0.188	**0.002**	−0.053	0.303
	Animal fats	−0.087	0.148	0.027	0.602
	Vegetable fats	−0.192	**0.001**	−0.120	**0.021**
	Saturated fats	−0.109	0.071	0.051	0.326
	Monounsaturated fats	−0.199	**0.001**	−0.097	0.061
	Polyunsaturated fats	−0.143	**0.017**	−0.080	0.120
	Linoleic acid	−0.137	**0.023**	−0.066	0.202
	Linolenic acid	−0.170	**0.004**	0.014	0.782
	Oleic acid	−0.205	**0.001**	−0.106	**0.041**
Proteins	Proteins	−0.166	**0.006**	−0.103	**0.047**
	Animal proteins	−0.163	**0.006**	−0.087	0.094
	Vegetable proteins	0.053	0.378	−0.111	**0.032**
Alcohol	Alcohol	0.176	**0.003**	0.148	**0.004**
Minerals	Calcium	−0.064	0.284	0.031	0.544
	Iron	−0.286	**0.000**	−0.058	0.263
	Phosphorus	−0.135	**0.025**	−0.046	0.372
	Potassium	−0.264	**0.000**	0.078	0.133
	Sodium	−0.026	0.672	−0.093	0.071
	Zinc	−0.177	**0.003**	−0.063	0.226
Vitamins	Vitamin A	−0.040	0.506	0.050	0.337
	Vitamin B1	−0.275	**0.000**	−0.080	0.122
	Vitamin B2	−0.107	0.076	−0.013	0.806
	Vitamin B3	−0.159	**0.008**	−0.022	0.677
	Vitamin B6	−0.311	**0.000**	−0.069	0.184
	Vitamin B9	−0.127	**0.034**	−0.044	0.399
	Vitamin C	−0.110	0.068	0.069	0.184
	Vitamin D	−0.201	**0.001**	0.048	0.357
	Vitamin E	−0.192	**0.001**	−0.133	**0.010**
	β-carotene	−0.131	**0.029**	0.001	0.981
n		93		125	

## Data Availability

Data are available from the corresponding author upon reasonable requests.

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
