# Peer review of "Gender Moderates the Associations Between Responsiveness to Alarming Oral Sensations, Depressive Symptoms, and Dietary Habits in Adolescents"

_nutrients, 2025, doi:10.3390/nu17101653_

Round 1

Reviewer 1 Report

Comments and Suggestions for Authors

Abstract:

The authors should explicitely describe a) study aim: instead of the sentence (lines 20, 21), please add the comcept why adolescents form a relevant target group, b) results: please add the name of the statistical analyses, significance and some concrete values, c) the design, whether it is a longitudinal study or not, and if the former one, retrospective or prospective.

Introduction:

1) First paragraph is well described.

2) Line 63: "Diet is increasingly recognized as a modifiable and cost-effective 63
factor related to depression" - this sentence is vague and needs further clarification.
Diet itself is not a factor in relation to depression - in what way? Certain dietary habits? Food contents? Cost-effective - in what way? Changes in dietary habits as lifestyle change for  depression management/prevention?

Methods:

"a large cohort of 232 healthy adolescents" - this sample does not cover a large cohort, in the contrary, seems like a pilot study. To make a valid cohort study, around 1000 participants would be apporpriate. In addition, we also need to know more about the selection of the participants. a) Location of data collection? b) Inclusion/exclusion critera? c) Is this a random/representtative sample? d) What is the frame of the study population from which the sample was selected? e) How was the sample size determined/calculated?

Scale: correct description.

Results:

Line 643: The title is not lengthy and not appropriate pls rephrase it.

Discussion

Line 755: I suggest to add a sperate section for Conclusion.

Author Response

Comments 1:

Abstract: The authors should explicitly describe a) study aim: instead of the sentence (lines 20, 21), please add the concept why adolescents form a relevant target group, b) results: please add the name of the statistical analyses, significance and some concrete values, c) the design, whether it is a longitudinal study or not, and if the former one, retrospective or prospective.

Response 1:

Thank you for pointing this out. The abstract has been revised as follow:

a)     We clarified the relevance of focusing on adolescents as a target group (Lines 20–23);

b)     We specified the study design as a cross-sectional study (Line 26);

c)     We added details to the results section, including the statistical analyses used (e.g., Wilcoxon Rank Sum Test, moderation analyses), significance levels, and relevant effect sizes (Lines 33–38).

Comments 2:

Introduction:

1) First paragraph is well described.

2) Line 63: "Diet is increasingly recognized as a modifiable and cost-effective 63
factor related to depression" - this sentence is vague and needs further clarification. Diet itself is not a factor in relation to depression - in what way? Certain dietary habits? Food contents? Cost-effective - in what way? Changes in dietary habits as lifestyle change for depression management/prevention?

Response 2:

We appreciate the positive feedback on the first paragraph of our introduction and agree with the reviewer that our previous sentences relating diet to depression may have seemed vague. We have now rephrased the paragraph (Lines 65-75) to enhance readability and explicitly emphasize that changes in dietary habits are increasingly recognized as one of the most promising modifiable risk factors for the prevention of depressive symptoms.

Comments 3:

Methods:

"a large cohort of 232 healthy adolescents" - this sample does not cover a large cohort, in the contrary, seems like a pilot study. To make a valid cohort study, around 1000 participants would be apporpriate. In addition, we also need to know more about the selection of the participants. a) Location of data collection? b) Inclusion/exclusion critera? c) Is this a random/representtative sample? d) What is the frame of the study population from which the sample was selected? e) How was the sample size determined/calculated?

 Scale: correct description.

Response 3:

Thank you for pointing this out. We acknowledge that the term “cohort” may have been misleading, as it often implies larger-scale epidemiological studies. In our manuscript, we intended the term to highlight shared characteristics among participants (e.g., age, health status, educational context), rather than to emphasize the size of the group. However, we fully understand the reviewer’s concern and have replaced “cohort” with alternatives such as “sample,” “group,” or “participants” throughout the revised manuscript to prevent confusion. Additionally, while the original manuscript included information on inclusion/exclusion criteria, and recruitment procedures (Section 2.1) or on the location of data collection (Section 2.2), we recognize that further context was necessary to improve clarity and completeness. Accordingly, the Participants section (Section 2.1; Ln 164–191) has been rephrased to provide a more comprehensive description. In details, Section 2.1 now includes the following information, expanding on previous content:

  1. Sample size determination (Ln 167-180): Given the novelty of our approach (e.g., integration of psychophysical and hedonic ratings, psychometric measures, dietary assessments, and the collection of salivary and tongue dorsum samples for metagenomic and/or genetic downstream applications in adolescents), no directly comparable studies were available. Therefore, sample size was estimated based on a prior study from our group involving 100 healthy young adults (aged 18-30) from the same geographic area [1]. In that study, we observed significant differences in responsiveness to bitterness and sourness evoked by a series of commercially-available foods (Cohen’s d = 0.402) between two groups with similar salivary microbial profiles. Using the pwr.t.test R function [2], we then estimated that a minimum of 198 participants would be required to detect this effect with 80% power at α = 0.05 (two-tailed). To account for deviations from normality and potential dropouts, we increased the target sample by 15%, resulting in a planned sample of 228 participants, which was later exceeded with the 232 adolescents involved in this work.
  2. Recruitment: Adolescents were recruited through promotional events targeting students and their parents (Ln 179-180). Participation was voluntary, and written informed consent from parents or legal guardians and assent from participants were obtained prior to data collection (Ln 188-189).
  3. Inclusion/exclusion criteria: No formal exclusion criteria were applied. However, participants self-reported no current diagnosis of major depressive disorder and no use of medications in the past six months that could affect mood or taste function (Ln 180-187).
  4. Sample representativeness: We believe the sample reasonably reflects typical students from the selected geographical and educational context and, to an extent, the broader Italian adolescent population within the framework of this study. Notably, our sample (IT-1) closely resembles a larger national cohort (IT-2) of 3.002 adolescents involved in a multicenter study on the 2017-2018 prevalence of mental health issues [3]. Specifically, our participants exhibited comparable rates of relevant depressive symptoms (IT-1: 21.6%; IT-2: 20.8%), with girls (IT-1: 36.1%; IT-2: 28.7%) showing threefold higher prevalence than boys (IT-1: 11.1%; IT-2: 10.4%) in both samples.
  5. Ethics: Our study was approved by the Research Ethics Committee of the University of Trento (n° prot. 2023-047, approved on 28/09/2023) and followed the principles of the Declaration of Helsinki (as amended in Fortaleza, Brazil, 2013).

Lastly, regarding the reviewer’s comment on the scale description, we are uncertain about the specific aspect being referred to. At this stage, we have opted not to amend the manuscript in this regard, as we believe the current descriptions of the measures used are appropriate and sufficiently detailed. Nevertheless, we would be willing to revise further should the reviewer provide additional clarification or specific guidance.

References

1.        Menghi, L.; Cliceri, D.; Fava, F.; Pindo, M.; Gaudioso, G.; Giacalone, D.; Gasperi, F. Salivary Microbial Profiles Associate with Responsiveness to Warning Oral Sensations and Dietary Intakes. Food Research International 2023, 171, 113072, doi:https://doi.org/10.1016/j.foodres.2023.113072.

2.        Champely, S.; Ekstrom, C.; Dalgaard, P.; Gill, J.; Weibelzahl, S.; Anandkumar, A.; Ford, C.; Volcic, R.; De Rosario, H.; De Rosario, M.H. Package ‘Pwr.’ R package version 2018, 1.

3.        Donato, F.; Triassi, M.; Loperto, I.; Maccaro, A.; Mentasti, S.; Crivillaro, F.; Elvetico, A.; Croce, E.; Raffetti, E. Symptoms of Mental Health Problems among Italian Adolescents in 2017–2018 School Year: A Multicenter Cross-Sectional Study. Environ Health Prev Med 2021, 26, doi:10.1186/S12199-021-00988-4.

Comments 4:

Results:

Line 643: The title is not lengthy and not appropriate pls rephrase it.

Response 4:

Thank you for pointing this out. We have revised the title of the subparagraph in: “4.2. Greater responsiveness to bitterness and astringency is associated with higher severity of depressive symptoms in adolescent girls, but not in boys” (Ln 651-652) to ensure clarity and consistency with its content. We hope this aligns with the reviewer’s expectations.

Comments 5:

Discussion

Line 755: I suggest to add a sperate section for Conclusion.

Response 5:

According to the reviewer’s suggestion, we have now included a separate section for conclusions (Ln 808-824).

Reviewer 2 Report

Comments and Suggestions for Authors

This study offers valuable insights into how gender influences the relationship between sensory responsiveness, depressive symptoms, and dietary habits in adolescents—a population often neglected in related research. However, despite its strengths, there are several limitations that should be acknowledged and more thoroughly discussed by the authors.

First, the cross-sectional nature of the study precludes any conclusions about causality. While associations are observed between sensory perception and depressive symptoms, it remains unclear whether sensory sensitivity contributes to mood disturbances, or if existing depressive traits alter sensory experiences and dietary behaviors. The reliance on self-reported dietary intake over a one-year period using the EPIC Food Frequency Questionnaire also introduces potential recall bias, especially among adolescents, whose eating patterns may be more variable and less accurately remembered.

Moreover, although the sample size is reasonable, it may still lack sufficient power to detect subtler interactions, particularly within subgroup analyses by gender. The use of highly specific sensory stimuli (grapefruit juice and chocolate pudding) may limit ecological validity, as these may not represent broader sensory or dietary experiences. Additionally, the complex psychological constructs involved—such as neuroticism, body dissatisfaction, and pickiness—are difficult to disentangle from depressive symptoms and may confound the observed relationships.

Given these methodological limitations, the discussion section would benefit from a more robust and explicit limitations subsection. The authors should elaborate on the implications of their design constraints, potential biases in dietary reporting, and the challenges of isolating individual psychological traits in observational studies. Acknowledging these issues would strengthen the credibility of their interpretations and guide future research toward more longitudinal, experimentally controlled designs.

Author Response

Comments 1:

This study offers valuable insights into how gender influences the relationship between sensory responsiveness, depressive symptoms, and dietary habits in adolescents—a population often neglected in related research. However, despite its strengths, there are several limitations that should be acknowledged and more thoroughly discussed by the authors.

First, the cross-sectional nature of the study precludes any conclusions about causality. While associations are observed between sensory perception and depressive symptoms, it remains unclear whether sensory sensitivity contributes to mood disturbances, or if existing depressive traits alter sensory experiences and dietary behaviors. The reliance on self-reported dietary intake over a one-year period using the EPIC Food Frequency Questionnaire also introduces potential recall bias, especially among adolescents, whose eating patterns may be more variable and less accurately remembered.

Moreover, although the sample size is reasonable, it may still lack sufficient power to detect subtler interactions, particularly within subgroup analyses by gender. The use of highly specific sensory stimuli (grapefruit juice and chocolate pudding) may limit ecological validity, as these may not represent broader sensory or dietary experiences. Additionally, the complex psychological constructs involved—such as neuroticism, body dissatisfaction, and pickiness—are difficult to disentangle from depressive symptoms and may confound the observed relationships.

Given these methodological limitations, the discussion section would benefit from a more robust and explicit limitations subsection. The authors should elaborate on the implications of their design constraints, potential biases in dietary reporting, and the challenges of isolating individual psychological traits in observational studies. Acknowledging these issues would strengthen the credibility of their interpretations and guide future research toward more longitudinal, experimentally controlled designs.

Response 1:

We thank the reviewer for the positive feedback and the thoughtful comment. We agree that strengthening the limitations section would enhance the credibility of our work and more clearly guide future research directions. Accordingly, we have added a dedicated Strengths and limitations section to the manuscript (Ln 762-807), in which we more thoroughly address the key methodological constraints of our study and discuss how these might be addressed in future investigations. We hope these revisions meet the reviewer’s expectations.
